# Influence of TiN Inclusions and Segregation Bands on the Mechanical Properties and Delayed Crack in Thick NM550 Wear-Resistant Steel

**DOI:** 10.3390/ma16175856

**Published:** 2023-08-26

**Authors:** Haoran Sun, Hegang Du, Keke Tong, Lihua Liu, Qiangjun Yan, Xiurong Zuo

**Affiliations:** 1Key Laboratory of Material Physics, Ministry of Education, School of Physics and Microelectronics, Zhengzhou University, Zhengzhou 450001, China; sunhaoranczy@163.com (H.S.); duhegangstudy@163.com (H.D.); tongkeke2022@163.com (K.T.); 2Nanjing Iron & Steel Co., Ltd., Nanjing 210035, China; njngllh@163.com (L.L.); yanqiangjun@njsteel.com.cn (Q.Y.)

**Keywords:** NM550 wear-resistant steel, segregation band, TiN inclusion, mechanical property, delayed crack

## Abstract

The formation mechanism of the delayed crack after flame cutting and mechanical properties in thick NM550 wear-resistant steel are studied by optical microscopy, scanning electron microscopy, energy dispersive spectroscopy, X-ray diffraction, and an electron backscattered diffractometer. The delayed crack is formed at the segregation zone (SZ) located in the center of the 65 mm thick steel plate. The strength of the non-segregation zone (NSZ) with a martensite microstructure is slightly higher than that of SZ with a mixture microstructure of martensite plus bainite, and the plasticity of NSZ is significantly better than that of SZ. There exists a more severe segregation in the SZ, and only a slight segregation in the NSZ. The average grain sizes of the segregation bands in the NSZ and SZ are 15.72 µm and 6.76 µm, respectively. The number density of TiN larger than 5 µm in the NSZ and SZ is 0.031 and 1.156 number/mm^2^, respectively. Therefore, a high hardness segregation band with fine grains and a high dislocation density, along with the large number of coarse TiN inclusions within it, results in delayed cracking. For TiN inclusions close to the crack, microvoids or microcracks around the TiN are formed, and the delayed crack will propagate along the edge of the TiN or through the TiN inclusions.

## 1. Introduction

Owing to the excellent combination of strength and wear resistance, thick NM550 wear-resistant steel is widely used in the mechanical equipment and engineering structures of mining, construction, metallurgy, and other industries [1]. However, delayed cracking in the flame cutting surface of the thick NM550 steel often happens after flame cutting, resulting in great harm with its subsequent use. For the safe use of thick NM550 steel, it is desirable to analyze the delayed cracking from the microscopic scale.

In the continuous casting of thick NM550 steel, alloying elements are enriched in the thickness center of the steel slab owing to the gradual decrease in cooling rates from the surface to the thickness center [2,3,4], which forms the segregation band after rolling. It has been found [5,6] that the delayed crack mostly initiates from the locally harder segregation bands in the center of the steel plate in the flame cutting surface, which fails to withstand the high residual stresses after flame cutting. Many studies [7,8,9] have found that hydrogen is typically very harmful for susceptible microstructures and the stress state, which results in hydrogen-induced delayed cracking in the steel plate after flame cutting. Many studies have also shown that the microalloying element Ti can improve the performance of low-alloy high-strength steel [1,10,11]. However, coarse TiN inclusions would impair the resistance to hydrogen embrittlement of the steel and result in hydrogen-induced delayed cracking after flame cutting due to inopportune Ti addition and process control [6,10,12]. In summary, the formation of a delayed crack in the steel plate is complicated, having relationships with both TiN inclusions and the segregation band. Wu et al. [13] and Liu et al. [6] compared the deformation mechanisms of TiN and CaO·(Al_2_O_3_)*_x_* inclusions during tensile tests and investigated the influence of the segregation band on delayed cracking propagation for NM500 and NM450 steel. However, interactions among TiN inclusions, segregation bands, and delayed cracking were not clear.

Therefore, the aim of this study was to investigate the interactions among TiN inclusions, segregation bands, and delayed cracking in NM550 steel. In order to investigate the influence of the segregation band on the fracture process, tensile testing was also conducted. The research presented in this paper can serve as a reference for resolving delayed cracking issues in the flame cutting of steel.

## 2. Materials and Methods

The material used in this study is an NM550 steel plate with a thickness of 65 mm, which was quenched and tempered after rolling. The chemical composition is shown in Table 1. The quenching and tempering temperatures are 900 °C and 160 °C, respectively. After tempering, the steel plate was flame cut with an oxyfuel propane gas. After placement for a certain time, a transverse delayed crack appeared on the steel plate. The delayed crack was observed on the longitudinal surface perpendicular to the flame cutting surface at the center of thickness by optical microscopy (OM; ZEISS, Axiolab 5, Oberkochen, Germany) and scanning electron microscopy (SEM; Quanta 250 FEG; FEI, Utah, UT, USA). The sectioned longitudinal surface of the crack and the sectioned transverse surface far from flame cutting surface were polished/etched with 4% Nital or 80 °C saturated picric acid solution, and then observed by OM and SEM.

A matrix sample with a size of 65 mm × 10 mm × 9 mm was cut for hardness tests by applying 2 kg force (XHD-2000TMSC; Vickers, Shanghai, China) on the transverse surface along the thickness direction of the steel plate. For the tensile testing, cylindrical specimens with a gauge length of 17.5 mm and diameter of 3.5 mm were prepared at the segregation zone (SZ) and the non–segregation zone (NSZ) far from the flame cutting surface according to the GB/T 228.1-2010 standard [14] (China National Standard, China, 2010). Room-temperature tensile testing was carried out at a speed of 10 mm per minute on a universal tensile testing machine (CMT5105; SANS, Shenzhen, China). The morphology of the tensile fracture surface was analyzed by SEM and an ultra-depth-of-field 3D microscope (DSX1000, Olympus Optical Corporation, Tokyo, Japan), and the inclusion compositions were qualitatively analyzed by an energy dispersive spectrometer (EDS; INCA–ENERGY, Oxford, UK). OM was used to measure the grain size and TiN inclusion size in the SZ and NSZ. The grain sizes were measured by using the 9–house method. A value of 32 mm² was selected for the TiN inclusion size measurement in the SZ and NSZ, respectively. Only a size of TiN larger than 0.5 µm was counted. It was found that the TiN quantity in the SZ and NSZ was 425 and 480, respectively.

Crystallographic analysis and microstructural characterization of SZ were carried out by a 20 kV field emission scanning electron microscope (FESEM; ZEISS, Oberkochen, Germany) integrated with an electron backscatter diffraction (EBSD) detector with a step of 0.3 µm. HKL Channel 5 software (Oxford, UK) was used for the data acquisition and post processing. The dislocation density was quantitatively analyzed used X-ray diffraction (XRD; PANalytical, Empyrean, Almelo, The Netherlands) with a Cu–Kα radiation source (λ = 1.5406 Å). The specimens were scanned over an angle range of 40–100°, with a step size of 0.013° and a speed of 0.49° min^−1^ after electropolishing, and the test lasted 124 min. The reflections of (110)α, (200)α, (211)α, and (220)α were measured. The X-ray data were post processed using the Jade software (MDI, JADE, V8.8). The base line of the XRD profile was removed and Kα2 elimination was performed. A broadening of the peaks in the X-ray profile occurred due to the dislocations generated during ratcheting deformation.

## 3. Results

### 3.1. Crack Analysis

The delayed crack located at the center of the plate thickness along the rolling direction (RD) is observed under OM (Figure 1b). The delayed crack initiated, propagated, and arrested repeatedly and would deflect during the propagation process. Figure 1c shows the crack morphology with picric acid corrosion under OM. It was found that the serious segregation bands centralized in the SZ at the thickness center of the steel plate, from which the delayed crack originated. The segregation bands contained more impurities and alloying elements than the surrounding matrix, resulting in the high hardness and low plasticity [5,6]. Figure 1d shows the morphology of the delay crack by SEM. Delayed crack arrest and re-initiating processes were noticed. At the same time, TiN was also found on the crack (Figure 1e). Hydrogen atoms were easily accumulated at the interface of the TiN inclusion and matrix, forming hydrogen molecules, which facilitated the initiation and propagation of the delayed cracks [9,12].

### 3.2. Mechanical Properties

Figure 2a shows the hardness along the thickness direction of the steel plate. The hardness values in the thickness range of 22–45 mm from the upper plate surface fluctuated greatly, which corresponded to the SZ, whereas the hardness fluctuation in the NSZ was gentle. It can be speculated that microstructure segregation caused the excessive fluctuation of hardness. Thus, further microstructure analysis was needed.

The zones of 3.5 mm between the red lines are the sampling position of the tensile samples in the NSZ and SZ (Figure 2a). The stress–strain curves are shown in Figure 2b. Table 2 shows the tensile properties of the samples in the NSZ and SZ. The elastic stages of the tensile curves were almost coincident, while the stress of the NSZ sample was larger than that of the SZ sample during plastic deformation. Interestingly, the tensile strength of the NSZ sample was slightly higher than that of SZ sample, whereas the area reduction of the NSZ sample was significantly better than that of the SZ sample. The work hardening exponent (*n*) of the NSZ sample and SZ sample was derived from the Hollomon’s equation [15]:(1)σ=s×(1+e)
(2)ε=ln (1+e)
(3)$$\sigma =k\varepsilon^{n}$$
where *s* is the engineering stress, *e* is the engineering strain, *σ* is the true stress, *ε* is the logarithmic strain (also referred as the true strain), and *k* is the Hollomon’s strength coefficient. It was calculated that *n*_sz_ and *n*_nsz_ were 0.23 and 0.21, respectively; the former was slightly higher than the latter. The *k* values of the SZ and NSZ were 4082 MPa and 3790 MPa, respectively.

### 3.3. Fractography

SEM was used to observe the fracture morphology of the tensile specimen in the SZ and NSZ, as shown in Figure 3. The tensile fracture surfaces of the two specimens were both composed of fiber zone and shear lip. However, distinct differences in the shapes of the fiber zones were noted: a circular shape with a diameter of 1605 µm in the NSZ sample (Figure 3a) and an elliptical shape with a long axis of 2523 µm and a short axis of 1788 µm in the SZ sample were found (Figure 3d). It was indicated that the microstructure of the NSZ sample was much more uniform than that of the SZ sample. The fracture surface was characterized by ductile dimples, tear ridges, secondary cracks, and cleavage facets in fiber zones. A higher proportion of cleavage facets with brittle characteristics on the fracture surface of the SZ specimen was found, compared with that of the NSZ. On the contrary, many deeper ductile dimples on the fracture surface of the NSZ specimen were found, compared with that of the SZ (Figure 3b,e). The shear lip region of the NSZ had smaller and shallower shear dimples than that of the SZ. However, there were some cleavage facets besides dimples in the shear lip region of the SZ (Figure 3c,f). In conclusion, it was indicated that the plasticity of the SZ was poorer than that of the NSZ [16].

Spherical CaO·(Al_2_O_3_)*_x_* inclusions indicated by the arrow in Figure 3h were found at the bottom of deep dimples, according to the EDS spectra shown in Figure 3j. The regular–shaped inclusions were identified as TiN inclusions containing small amounts of Nb by EDS analysis (Figure 3k,l), which are indicated by arrows in Figure 3g,i. TiN inclusions with a small size caused deep dimples (Figure 3g). In contrast, TiN inclusions with a large size caused brittle cleavage (Figure 3i), which was more harmful to plasticity than the small-size TiN and spherical CaO·(Al_2_O_3_)*_x_* inclusions [17,18].

### 3.4. Microstructure

#### 3.4.1. Microstructure of the NSZ and SZ

Microstructure of the NSZ and SZ under OM is shown in Figure 4. The microstructure of the NSZ was uniform lath martensite (Figure 4a), whose hardness value was relatively uniform, with an average value of about 590 HV (Figure 2a). However, the enrichment of elements (C, Cr, Mn and Mo) existed in the segregation band in the SZ by EDS analysis (Figure 4c,d) [2,3], forming a microstructure of martensite in segregation band and bainite plus martensite in surrounding matrix (Figure 4b). As a result, the hardness of the SZ fluctuated greatly. The hardness values of martensite in segregation band and bainite plus martensite in matrix of the SZ were 717 HV and 515 HV, respectively, and the former was 202 HV higher than the latter (Figure 4b). In addition, the NSZ was composed of martensite and the SZ was a mixture of martensite and bainite, resulting in the strength of NSZ slightly higher than that of the SZ [19], as shown in Figure 2b. Coarse TiN inclusions was also found in the segregation band in the SZ (Figure 4b).

#### 3.4.2. Grain Size Distribution in the NSZ and SZ

Morphologies of the NSZ and SZ corroded by Picric acid under OM are shown in Figure 5. It was shown that microstructure in NSZ displayed slight segregation bands, whereas microstructure in the SZ had severe segregation. In the NSZ, the average grain sizes of segregation bands, matrix and whole region were 15.72 µm, 26.47 µm, and 22.12 µm, respectively, which were lager than those in the SZ (6.76 µm, 25.53 µm, and 15.26 µm, respectively), implying obvious grain size refining in segregation band located in the SZ. In addition, the standard deviation of grain size in the SZ and NSZ was 12.55 µm and 10.32 µm, respectively, which displayed more uniform grain in the NSZ. Figure 5d shows the grain size distribution of the NSZ and SZ. It was evident that the grain size of the NSZ and SZ obeyed a normal distribution, and the grain size of the former was larger but more uniform than that of the latter. Segregation bands with fine grains displayed high hardness, which had high HIC susceptibility [8].

#### 3.4.3. TiN Inclusion Size Distribution in the NSZ and SZ

Figure 6 shows the homogeneous and heterogeneous nucleated TiN inclusions and size distribution in the NSZ and SZ. In this experiment, the statistical limit of TiN size was 0.5 µm under OM. The size and number densities of TiN inclusions in the NSZ and SZ were different. The average and maximum sizes of TiN inclusions in the NSZ and SZ were 2.44 µm and 3.05 µm, 5.24 µm, and 14.47 µm, respectively. The size of TiN inclusions in the SZ was larger than that of the NSZ, which meant that TiN in the SZ at the thickness center of the plate would be more detrimental to the properties of steel than that in the NSZ near the surface of the plate.

The TiN inclusions larger than 5 µm were related to cleavage facets [10]. The number density of TiN larger than 5 µm in the NSZ and SZ was 0.031 and 1.156 number/mm^2^, respectively. Number density of the latter was much more than that of the former, which implied that there were more potential initiators of cleavage facets in the fracture surface of the SZ than that of the NSZ [10]. As a result, the poor elongation and area reduction observed in the SZ samples can be primarily attributed to the TiN inclusions larger than 5 μm.

It was found that coarse TiN inclusions were more prone to form at the thickness center of the steel plate, which almost all existed in segregation bands (Figure 4b and Figure 5c). This was because the continuous casting billet was about 350 mm in thickness; thus, the cooling rate of molten steel in the thickness center of the steel billet was much lower than that in the surface zone, which led to alloy elements including Ti diverging towards the thickness center in the steel [2,3,4].

#### 3.4.4. EBSD Analysis of the SZ

The kernel average misorientation (KAM) maps of the SZ are shown in Figure 7. The KAM map was used to approximately indicate the dislocation density. There are more orange and red areas in the segregation band than in the surrounding matrix in Figure 7a, demonstrating the higher local plastic strain and the stored energy in the segregation band [9]. Figure 7b,c shows that most of the orange and red areas with a high strain were located at the plate boundaries of martensite, which were prone to inducing HIC microcracks. The KAM values of the segregation band were relatively higher (KAM_ave_ = 0.82°, KAM_peak_ = 0.65°) than those of the matrix (KAM_ave_ = 0.74°, KAM_peak_ = 0.55°); thus, the strain gradients were larger at the interface between them [20], where the trapping of hydrogen atoms promoted crack initiation and propagation.

Figure 8a,b shows the distribution map of the deformed grains, recrystallized grains, and substructures represented with red, blue, and yellow colors, respectively. Obviously, there were more deformed grains with a higher dislocation density and more energy stored in the segregation bands than in the matrix, which allowed the crack to spread more easily in the segregation band [21]. Therefore, dislocations strengthening in the segregation band resulted in a higher hardness those that in the surrounding matrix (Figure 4b).

The grain boundary distribution is shown in Figure 8c,d, where the black line represents the misorientation angles in the range of 15° ≤ *θ* ≤ 50° and the red line represents those of *θ* > 50°. Misorientation angles greater than 50° occupied 38.8% and 30.5% in the matrix and segregation band, respectively. Grain boundaries larger than 50° can consume more energy and effectively prevent crack propagation [21]. Therefore, the crack was easier to arrest in the matrix than in the segregation band.

#### 3.4.5. Dislocation Density Analysis

XRD analysis was used to quantify the difference in the dislocation density in the NSZ and SZ, as shown in Figure 9. The dislocation density was determined by the modified Williamson–Hall (MWH) method [22], as follows:(4)$$\Delta K=\frac{0.9}{d}+\frac{1}{2}\;\pi M²b²\sqrt{\rho }\;K²\bar{C}$$
where Δ*K* is the full width at half maximum of the diffraction peak, *K* = 2sin*θ*/*λ*, *θ* is the diffraction angle, *λ* = 0.15405 nm, *b* is the Burgers vector (0.248 nm), and *d* is the crystallite size. Furthermore, M is a constant (herein, the M value is 1.4), and $\bar{C}$ is the average contrast factor [22]. The result shows that the dislocation density of the NSZ (5.42 × 10^14^ m^−2^) was slightly higher than that of the SZ (2.05 × 10^14^ m^−2^), as shown in Figure 9c. This was because the microstructure of the NSZ with slight segregation was mainly martensite with a high dislocation density. However, a mixture of bainite and martensite in the SZ with severe segregation, in which the dislocation density of bainite was lower than that of martensite, resulted in a slightly lower dislocation density in the SZ than in the NSZ [23].

## 4. Discussion

### 4.1. Effect of Microstructure on Mechanical Properties

A cross section of 65 mm thick steel plate was divided into the SZ and NSZ according to the hardness fluctuation and segregation band distribution in the thickness direction (Figure 1a, Figure 2a, Figure 4 and Figure 5). The microstructure in the NSZ displayed slight segregation bands, whereas the microstructure in the SZ exhibited severe segregation (Figure 5a,b). The microstructure of the NSZ was a uniform lath martensite (Figure 4a). However, a mixture microstructure of martensite in the segregation band and bainite plus martensite in the surrounding matrix was formed in the SZ (Figure 4b). As a result, the hardness values fluctuated greatly in the SZ located at 22–45 mm thickness from the upper plate surface, whereas the hardness fluctuation was gentle in the NSZ, as shown in Figure 2a.

The area reduction of the NSZ tensile sample was significantly better than that of the SZ sample, despite the similar strength. The number density of TiN inclusions larger than 5 µm in the NSZ and SZ was 0.031 and 1.156 number/mm^2^, respectively. This was a result of the coarse TiN inclusions, which almost all existed in the segregation bands (Figure 4b and Figure 5c); these inclusions were more prone to form in the SZ at the thickness center of the steel plate, due to the Ti and other element enrichment towards the thickness center of the steel slab during the continuous casting process [2,3,4]. In general, there were more and larger TiN inclusions in segregation bands located in the SZ, which would result in the acceleration of crack formation and the deterioration of plasticity.

There existed severe segregation in the SZ; only slight segregation existed in the NSZ. The average grain sizes of the segregation bands in the NSZ and SZ were 15.72 µm and 6.76 µm, respectively (Figure 5). Briefly, severe segregation bands with fine grains and a high dislocation density (Figure 7 and Figure 8), along with the large number of coarse TiN inclusions within segregation bands (Figure 6), resulted in poor plasticity in the SZ.

### 4.2. Effect of Segregation Band on Tensile Fracture Shape

To analyze the inhomogeneity of the tensile fracture surface of the SZ sample, the influence of the segregation band on necking during the tensile process was studied, as shown in Figure 10.

Once necking occurs in the tensile process, the specimen would be subjected to multiaxial stress in the necking region before fracture. Point B was not only subjected to axial stress (*S*_z_), but also radial stress (*S*_r_) and tangential stress (*S*_t_), as shown in Figure 10c. Masanobu Murata [24] used the Bridgman’s formula to analyze the stress in the necking cross section of the sample, and described the relationship between *S*_r_, *S*_t_, *S*_z_, and flow stress (*S*_flow_). The equations are as follows:(5)$$S_{\mathrm{flow}}=\frac{1}{\left(1+2\frac{R}{a}\right)\;\ln\;\left(1+\frac{a}{2R}\right)}\sigma$$
(6)$$S_{r}=S_{t}=S_{\mathrm{flow}}\times \ln\;(\frac{a^{2}+2aR-r^{2}}{2aR})$$
(7)$$S_{z}=S_{\mathrm{flow}}\times \{1+\ln\;\left(\frac{a^{2}+2aR-r^{2}}{2aR}\right)\}$$
where *σ* is the true stress given by Equation (1), *a* is the radius of the necking minimum cross section (instantaneous), *R* is the radius of curvature at the bottom of the necking area, and *r* is the distance between the point B and the center axial of the specimen. The transverse stress at point B (including *S*_r_ and *S*_t_) increased with the decrease in *r*, reaching its maximum value at the central axis of the specimen.

After necking, the NSZ specimen showed isotropic deformation resistance during the tensile process due to its homogeneous microstructure, which was mainly martensite (Figure 4a). Therefore, the fracture surface shape was approximately circular, as shown in Figure 3a and Figure 10d. While the SZ specimen had uneven microstructure, including the segregation bands with martensite and the surrounding matrix with bainite plus martensite (Figure 4b); therefore, its deformation resistance showed anisotropy during the tensile process [25].

As discussed in Figure 4b, the hardness difference value between the segregation bands and the nearby matrix was more than 200 HV. Hence, the segregation bands were considered to be the hard domain and the matrix was considered to be the soft domain [26]. After necking, the soft domains of the matrix would start plastic deformation first, while the hard domains of the segregation bands remained elastic, which would cause back stress in the soft domains to increase the overall yield strength of the SZ. Thus, there was a slight difference in yield strength between the NSZ and SZ samples. And the back stress in the soft domain produced extra work hardening, resulting in an *n* and *k* value of the SZ slightly higher than that of the NSZ samples.

In the SZ, segregation bands with fine martensite were too hard to deform, resulting in the uniform distribution of small strain during the tensile process, whereas the soft matrix located at the interband region with an inhomogeneous distribution of microstructure (Figure 4b) showed a large strain [20,25]. It was this kind of inhomogeneous strain distribution that induced the formation of the crack along the transverse interface between the matrix and the segregation bands (Figure 10f). Owing to the increase in the transverse stress with the decrease in the distance to the central axis, as shown in Equation (6), the crack was much more prone to initiation at the location near the central axis with a maximum *S*_r_ and *S*_t_, followed by accelerating propagation along the interface of the matrix/segregation bands (Crack A) as opposed to initiation perpendicular to the interface (Crack B), which eventually merged and led to the fracture. However, cracks perpendicular to the interface could propagate across segregation bands, which was hindered by the nearby matrix with a low KAM value and a higher proportion of misorientation angles greater than 50° (Figure 7 and Figure 8), changed the propagation direction, and eventually arrested in the matrix (Figure 10f). As a result, the elliptical-shaped fracture surface of the SZ was formed, as shown in Figure 3d and Figure 10e. However, the crack initiation from coarse TiN inclusions within segregation bands also occurred and propagated to the interface of the matrix/segregation bands, which could be arrested by the matrix with a low hardness.

### 4.3. Interactions of TiN Inclusions and Delayed Crack

In order to analyze the interactions of TiN inclusions and the delayed crack, various positions of TiN inclusions relative to delayed cracks are shown in Figure 11. Schematic diagrams are shown in Figure 12.

Generally, when TiN inclusions were far away from the delayed crack, the interactions of TiN inclusions and the delayed crack were not obvious. However, for TiN inclusions close to the crack, subjected to the strain field of the delayed crack tip, microvoids or microcracks formed around TiN, and the delayed crack propagated along the edge of the TiN inclusions or through the TiN inclusions.

Due to the large difference in the thermal expansion coefficient, there was be a high thermal stress between the steel matrix and TiN inclusions, which produced a certain plastic zone with a high strain field [27]. At the same time, when the delayed crack propagated forward approaching the TiN in front of it, TiN was also affected by the strain fields of the main crack. The overlay of the two kinds of strain field resulted in microvoids or microcracks forming around the TiN. The hydrogen in the lattice moved to the corners of the TiN or delayed crack tip by stress-induced diffusion, leading to a microcrack appearing and delayed crack propagation [8,12].

The interactions between TiN inclusions without nuclei and the delayed crack are shown in the blue rectangles of Figure 11 and Figure 12a, and include three critical steps: (ⅰ) The delayed crack firstly propagated forward, approaching the TiN in front of it. (ⅱ) TiN was then affected by the strain field of the main crack, forming the microvoids and microcracks around TiN, which were generally located in segregation bands. (ⅲ) Finally, the main crack propagated across the TiN/matrix boundary and continued to extend forward.

The interactions between TiN inclusions with heterogeneous nucleation and the delayed crack are shown in black rectangle of Figure 11b and Figure 12b, which included three critical steps: (ⅰ) The delayed crack firstly propagated forward, approaching the TiN in front of it. (ⅱ) TiN was then affected by the strain fields of the main crack, resulting in the breaking of TiN from the oxide core with low hardness [28]. (ⅲ) Finally, the main crack propagated through the TiN and continued to extend forward.

## 5. Conclusions

Delayed cracking appears following placement for a certain time after flame cutting for the NM550 steel plate. The delayed crack arrest and re-initiation repeatedly occur with obvious delay characteristics.The area reduction in the NSZ tensile sample was significantly better than that of the SZ sample, despite the similar strength. The tensile fracture surface shape is approximately circular in the NSZ specimen due to isotropic deformation resistance with the homogeneous martensite microstructure, while an elliptical-shaped fracture surface in the SZ specimen is found due to the uneven microstructure including the segregation bands with martensite and the surrounding matrix with bainite plus martensite.The average and maximum sizes of TiN inclusions in the NSZ and SZ were 2.44 µm and 3.05 µm, and 5.24 µm and 14.47 µm, respectively, of which TiN inclusions in the SZ were larger than those in the NSZ. The number density of TiN inclusions larger than 5 µm in the NSZ and SZ was 0.031 and 1.156 number/mm^2^, respectively, of which the latter was much higher than that of the former. Therefore, it can be concluded that TiN in the SZ at the thickness center of the plate would be more detrimental to the plasticity of steel than that in the NSZ near the surface of the plate.There exists a more severe segregation in the SZ; only slight segregation exists in the NSZ. The hardness difference value between the segregation bands and the surrounding matrix was more than 200 HV in the SZ. The average grain sizes of the segregation bands in the NSZ and SZ were 15.72 µm and 6.76 µm, respectively. Briefly, severe segregation bands with fine grains and a high dislocation density display high hardness, along with a large number of coarse TiN inclusions within the segregation band, resulting in the poor plasticity and high HIC susceptibility in the SZ.For TiN inclusions far away from the delayed crack, the interactions of TiN inclusions and the delayed crack were not obvious. However, for TiN inclusions close to the crack, subjected to the strain field of a delayed crack tip, microvoids or microcracks around TiN formed, and the delayed crack propagated along the edge of the TiN inclusions or through the TiN inclusions.

## Figures and Tables

**Figure 1 materials-16-05856-f001:**
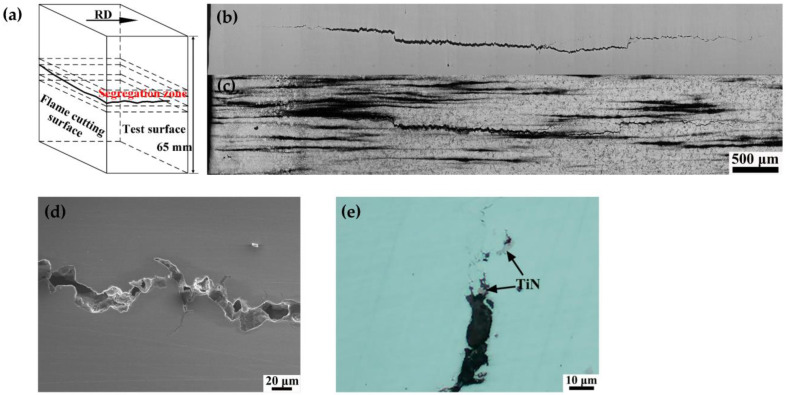
(**a**) Crack schematic, (**b**) crack morphology without corrosion by OM, (**c**) crack morphology with picric acid corrosion by OM, (**d**) crack morphology by SEM, (**e**) TiN inclusions located at the delayed crack by OM.

**Figure 2 materials-16-05856-f002:**
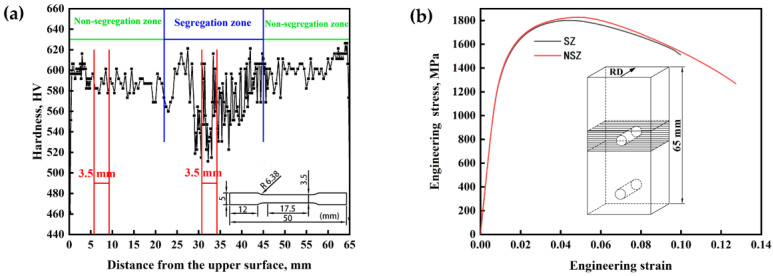
(**a**) Hardness curve in thickness direction, (**b**) stress–strain curves of the SZ sample and the NSZ sample along the rolling direction.

**Figure 3 materials-16-05856-f003:**
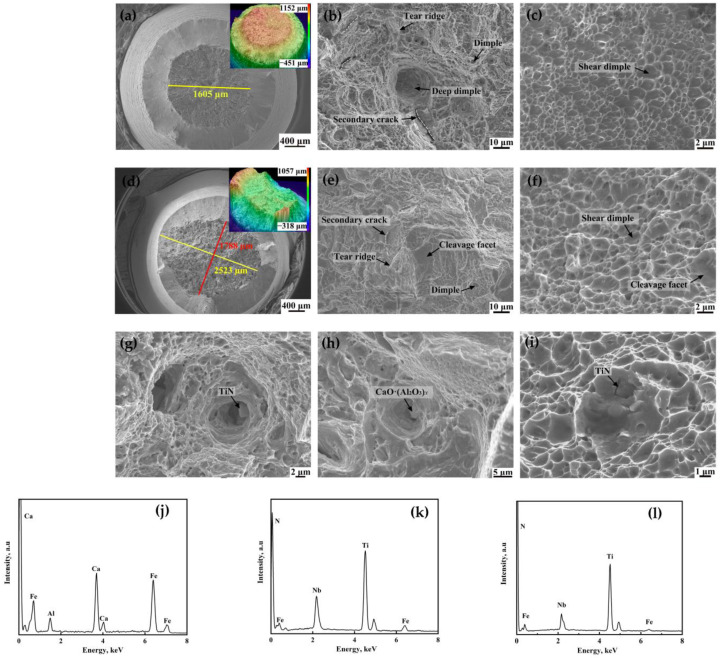
Tensile fracture surfaces of the specimens after tensile testing. (**a**) Macro fracture surface of the NSZ sample, (**b**) fiber zone of the NSZ sample, (**c**) shear-lip zone of the NSZ sample, (**d**) macro fracture surface of the SZ sample, (**e**) fiber zone of the SZ sample, (**f**) shear-lip zone of the SZ sample, (**g**) TiN inclusion of the NSZ sample, (**h**) CaO·(Al_2_O_3_)*_x_* inclusion of the SZ sample, (**i**) TiN inclusion of the SZ sample, and (**j**–**l**) EDS of the inclusions.

**Figure 4 materials-16-05856-f004:**
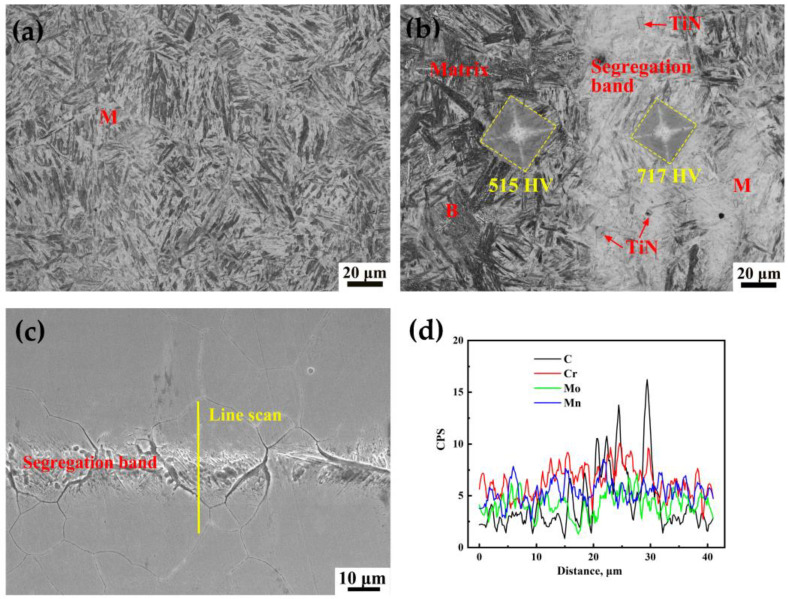
Microstructure under OM of (**a**) the NSZ and (**b**) the SZ. (**c**,**d**) EDS line scan of the segregation band, containing bainite (B) and martensite (M).

**Figure 5 materials-16-05856-f005:**
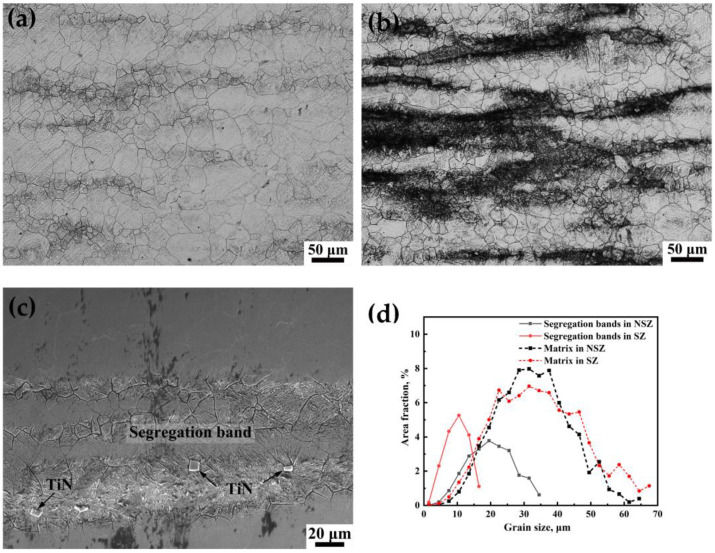
Grain morphology by picric acid corrosion of (**a**) the NSZ under OM, (**b**) the SZ under OM, (**c**) the SZ under SEM, and (**d**) the grain size distribution of the NSZ and SZ.

**Figure 6 materials-16-05856-f006:**
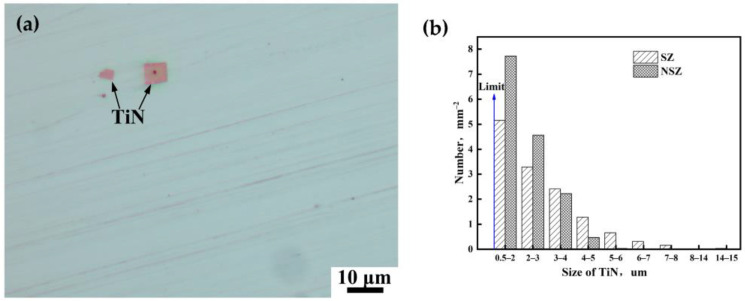
(**a**) TiN inclusions in SZ under OM, and (**b**) the size distribution of TiN inclusions in the NSZ and SZ.

**Figure 7 materials-16-05856-f007:**
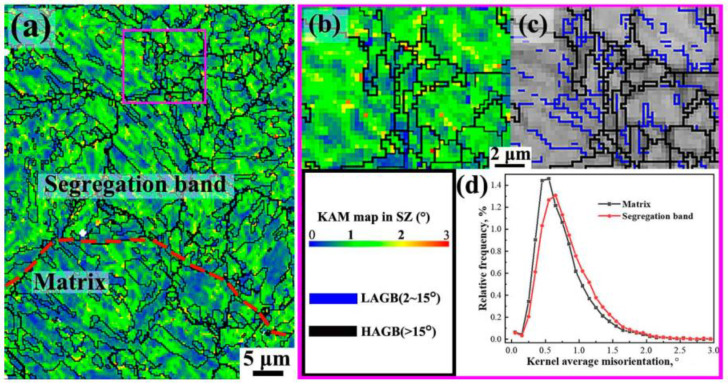
(**a**) Kernel average misorientation (KAM) map for segregation band and matrix, (**b**) KAM map for segregation band, (**c**) grain boundary misorientation map of segregation band, and (**d**) distribution of local misorientation.

**Figure 8 materials-16-05856-f008:**
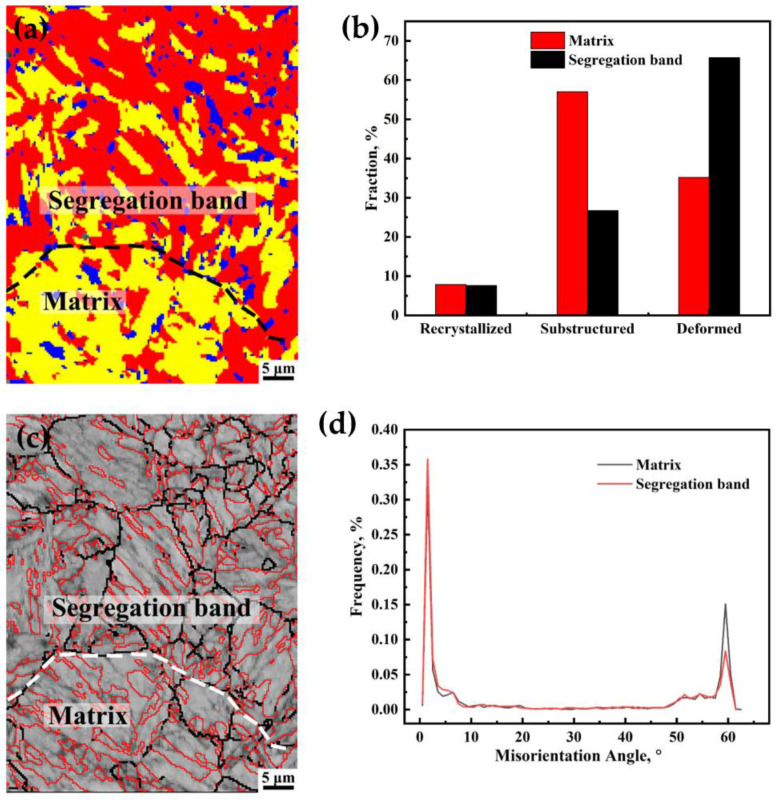
(**a**) Distribution map of deformed grains, recrystallized grains, and substructures of segregation band and matrix, (**b**) volume fractions of recrystallized, substructures, and deformed grains. (**c**) grain boundary misorientation map of the segregation band and matrix, and (**d**) distribution of the grain boundary misorientation.

**Figure 9 materials-16-05856-f009:**
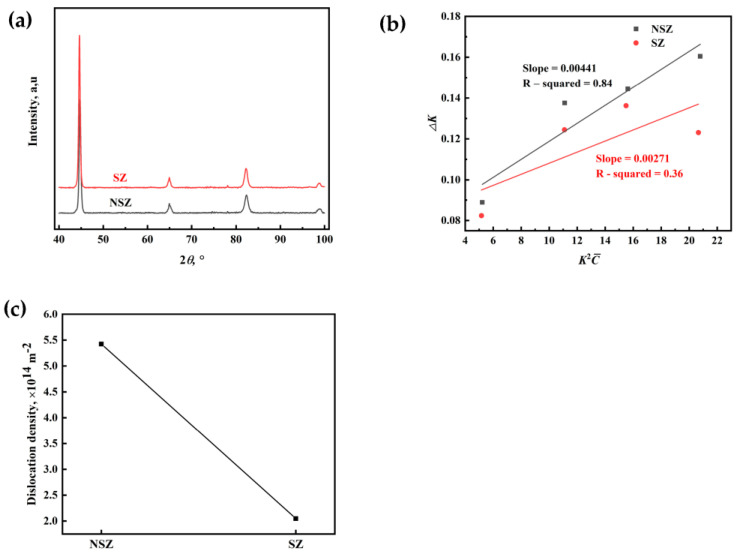
(**a**) X-ray diffraction profiles of the NSZ and SZ, (**b**) the fitted lines of Δ*K* vs. $K\bar{C}$ based on the modified Williamson-Hall method, and (**c**) the calculated dislocation density of the NSZ and SZ.

**Figure 10 materials-16-05856-f010:**
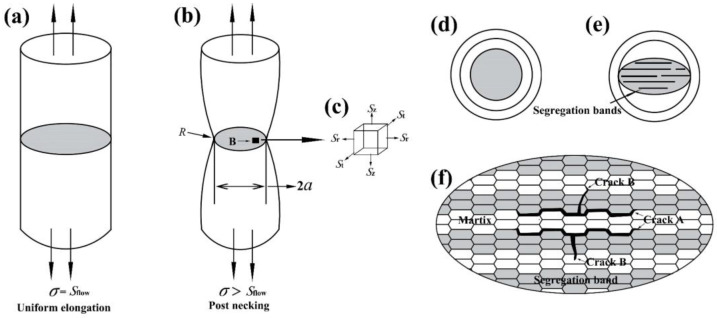
Schematic diagram of tensile specimens’ (**a**) uniform deformation, (**b**) post necking, (**c**) the stress distribution in the necking area, (**d**) circular fracture surface of the NSZ specimen, (**e**) elliptic fracture surface of the SZ specimen, and (**f**) schematic diagram of fracture surface of the SZ specimen.

**Figure 11 materials-16-05856-f011:**
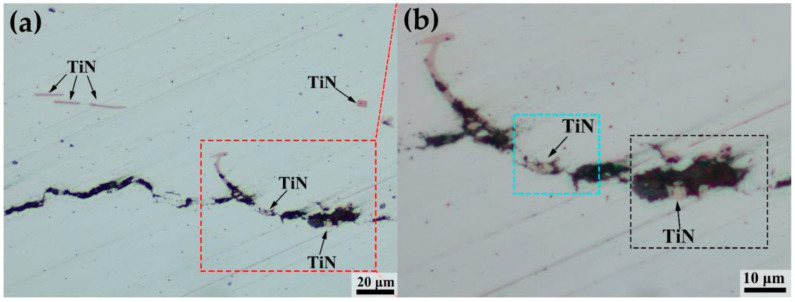
Position of TiN inclusions with respect to the crack of (**a**,**b**).

**Figure 12 materials-16-05856-f012:**
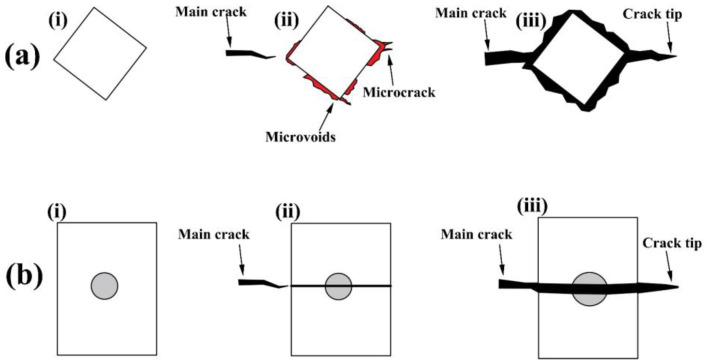
(**a**) Delayed crack across the interface of the matrix/TiN inclusion, and (**b**) delayed crack through the TiN inclusion.

**Table 1 materials-16-05856-t001:** Chemical compositions of NM550 wear–resistant steel (mass%).

C	Mn	P	S	Si	Ni + Mo + Cr	Nb + Ti
0.33	0.79	0.0084	0.0013	0.25	1.91	0.032

**Table 2 materials-16-05856-t002:** Tensile properties of the NM550 steel specimens in the SZ and NSZ.

Location	Tensile Strength (MPa)	Yield Strength (MPa)	Elongation (%)	Area Reduction (%)
SZ	1800	1340	14.3	28.5
NSZ	1830	1365	16.6	43.1

## Data Availability

Data are contained within the article.

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
