# Peer review of "Influence of TiN Inclusions and Segregation Bands on the Mechanical Properties and Delayed Crack in Thick NM550 Wear-Resistant Steel"

_materials, 2023, doi:10.3390/ma16175856_

Round 1

Reviewer 1 Report

The article considers the initiation of cracks and mechanisms of cracking in wear-resistant steel NM550, and the authors used a fairly large number of different methods of analysis to characterize the observed effects, the combination of which allowed them to obtain a number of unique results. In general, the presented article has a high level of scientific novelty and practical significance, and the results themselves are worthy of being accepted for publication after the authors provide clarifications to a number of questions that a reviewer has when analyzing this article. It should also be noted that the article corresponds to the subject of the declared journal and in the future, if accepted, will be able to make a significant contribution to the development of this area of research.

1. The results of X-ray diffraction analysis should be given with a detailed description of the observed changes (if any), as well as a comparison of reflection intensities in order to determine the possible occurrence of textural effects.

2. The results of the energy dispersive analysis shown in Figure 3 should be given with a reflection of the places where they were obtained from.

3. The authors should explain such a large spread of hardness values in depth, as well as how exactly they determined the segregation zone in such a large spread.

4. According to the presented images, the inclusion of titanium nitride grains is observed near the cracks, the authors should explain the reasons for the observed inclusions, as well as their effect on the containment or, conversely, the acceleration of crack formation processes.

5. The authors describe the effect of grain size reduction and an increase in dislocation density as one of the mechanisms for increasing crack resistance, which in general takes place and is confirmed by a large number of various experimental works. However, in the case of the presented data, more attention should be paid to the dynamics of grain size changes and the ratio of large and small grains in the samples, as well as various inclusions that can have a similar strengthening effect.

Reviewer 2 Report

The manuscript presents the results of a study focused on the formation mechanism of delayed crack after flame cutting. The work is original and interesting, however some aspects must be improved before it can be accepted for publication.

3.2. Mechanical Properties: Figure 2 b) shows the sampling positions of probes used in tensile tests, however the manuscript does not report details about the process used to cut the probes. This is important because the procedure may alter the material, in particular possible residual stresses present in segregation zones can be released.

3.4.5. Dislocation density analysis: The dislocation density was determined by the modified Williamson–Hall method that requires great accuracy in the measurement of XRD peak profiles. The experimental conditions (step size of 0.02° and speed of 1° / minute) do not provide sufficient precision in determining the full width at half maximum of the diffraction peaks. If ΔK has been determined by XRD spectra displayed in Figure 9 a) its value is affected by an enormous error. Moreover, the authors do not report any information about instrumental peak broadening, Kα2 stripping and background subtraction.

4. Discussion: the discussion should involve all the experimental results. In this case some of them (e.g. EBSD data) are not adequately considered. The discussion must be improved by taking into account all the data achieved by experiments and describing the way they are related.

English should be improved

English should be improved

Round 2

Reviewer 1 Report

The authors answered all the questions, the article can be accepted for publication.

Reviewer 2 Report

Authors modified the manuscript according to reviewer's suggestions and criticism thus it can be published in present form.